# High-order functional redundancy in ageing explained via alterations in the connectome in a whole-brain model

Marilyn Gatica[1,2], Fernando E. Rosas[3,4,5,6], Pedro A. M. Mediano[7,8], Ibai Diez[9,10,11], Stephan P. Swinnen[12,13], Patricio Orio[1,14], Rodrigo Cofré[15,16]☯*, Jesus M. Cortes[17,18,19]☯*

**1** Centro Interdisciplinario de Neurociencia de Valparaíso, Universidad de Valparaíso, Valparaíso, Chile, **2** Biomedical Research Doctorate Program, University of the Basque Country (UPV/EHU), Leioa, Spain, **3** Centre for Psychedelic Research, Department of Brain Science, Imperial College London, London, United Kingdom, **4** Data Science Institute, Imperial College London, London, United Kingdom, **5** Center for Complexity Science, Imperial College London, London, United Kingdom, **6** Department of Informatics, University of Sussex, Brighton, United Kingdom, **7** Department of Psychology, University of Cambridge, Cambridge, United Kingdom, **8** Department of Psychology, Queen Mary University of London, London, United Kingdom, **9** Department of Radiology, Division of Nuclear Medicine and Molecular Imaging, Massachusetts General Hospital and Harvard Medical School, Boston, Massachusetts, United States of America, **10** Gordon Center for Medical Imaging, Department of Radiology, Massachusetts General Hospital and Harvard Medical School, Boston, Massachusetts, United States of America, **11** Athinoula A. Martinos Center for Biomedical Imaging, Massachusetts General Hospital, Harvard Medical School, Boston, Massachusetts, United States of America, **12** Research Center for Movement Control and Neuroplasticity, Department of Movement Sciences, KU Leuven, Leuven, Belgium, **13** KU Leuven Brain Institute (LBI), KU Leuven, Leuven, Belgium, **14** Instituto de Neurociencias, Facultad de Ciencias, Universidad de Valparaíso, Valparaíso, Chile, **15** CIMFAV-Ingemat, Facultad de Ingeniería, Universidad de Valparaíso, Valparaíso, Chile, **16** Department of Integrative and Computational Neuroscience, Paris-Saclay Institute of Neuroscience, Centre National de la Recherche Scientifique, Gif-sur-Yvette, France, **17** Neuroimaging Lab, Biocruces-Bizkaia Health Research Institute, Barakaldo, Spain, **18** IKERBASQUE: The Basque Foundation for Science, Bilbao, Spain, **19** Department of Cell Biology and Histology, University of the Basque Country, Leioa, Spain

☯ These authors contributed equally to this work.
* rodrigo.cofre@uv.cl (RC); jesus.m.cortes@gmail.com (JMC)

**Data Availability Statement:** The BOLD signals and connectomes used in this study are available at https://github.com/brincolab/High-Order-interactions/tree/master/dataset.

## Abstract

The human brain generates a rich repertoire of spatio-temporal activity patterns, which support a wide variety of motor and cognitive functions. These patterns of activity change with age in a multi-factorial manner. One of these factors is the variations in the brain's connectomics that occurs along the lifespan. However, the precise relationship between high-order functional interactions and connnectomics, as well as their variations with age are largely unknown, in part due to the absence of mechanistic models that can efficiently map brain connnectomics to functional connectivity in aging. To investigate this issue, we have built a neurobiologically-realistic whole-brain computational model using both anatomical and functional MRI data from 161 participants ranging from 10 to 80 years old. We show that the differences in high-order functional interactions between age groups can be largely explained by variations in the connectome. Based on this finding, we propose a simple neurodegeneration model that is representative of normal physiological aging. As such, when applied to connectomes of young participant it reproduces the age-variations that occur in the high-order structure of the functional data. Overall, these results begin to disentangle the

**Funding:** MG was partially supported by CONICYT-PFCHA/ Doctorado Nacional/2019-21190577. FR acknowledges the support of the Ad Astra Chandaria Foundation. PM was funded by the Wellcome Trust (grant no. 210920/Z/18/Z). SPS was supported by the FWO Research Foundation Flanders (G089818N), the Excellence of Science funding competition (EOS; 30446199) and the KU Leuven Special Research Fund (grant C16/15/070). PO is funded by Fondecyt Regular grant 1211750 and ANID-Basal Project FB0008. RC acknowledges financial support from the Human Brain Project, H2020-945539. JMC acknowledges financial support from Ikerbasque (The Basque Foundation for Science) and from the Ministerio Economia, Industria y Competitividad (Spain) and FEDER (grant DPI2016-79874- R), as well as from the Department of Economic Development and Infrastructures of the Basque Country (Elkartek Program, KK-2018/00032, KK-2018/00090, and KK-2021/00009). The Centro Interdisciplinario de Neurociencia de Valparaiso (CINV) is supported by Grant ACE210014 (ANID). The funders had no role in study design, data collection and analysis, decision to publish, or preparation of the manuscript.

**Competing interests:** The authors have declared that no competing interests exist.

mechanisms by which structural changes in the connectome lead to functional differences in the ageing brain. Our model can also serve as a starting point for modeling more complex forms of pathological ageing or cognitive deficits.

## Author summary

Modern neuroimaging techniques allow us to study how the human brain's anatomical architecture (a.k.a. structural connectome) changes under different conditions or interventions. Recently, using functional neuroimaging data, we have shown that complex patterns of interactions between brain areas change along the lifespan, exhibiting increased redundant interactions in the older population. However, the mechanisms that underlie these functional differences are still unclear. Here, we extended this work and hypothesized that the variations of functional patterns can be explained by the dynamics of the brain's anatomical networks, which are known to degenerate as we age. To test this hypothesis, we implemented a whole-brain model of neuronal activity, where different brain regions are anatomically wired using real connectomes from 161 participants with ages ranging from 10 to 80 years old. Analyzing different functional aspects of brain activity when varying the empirical connectomes, we show that the increased redundancy found in the older group can indeed be explained by precise rules affecting anatomical connectivity, thus emphasizing the critical role that the brain connectome plays for shaping complex functional interactions and the efficiency in the global communication of the human brain.

## Introduction

Advancing the neuroscience of ageing is highly relevant from a socio-economic perspective as the population of older adults is rising dramatically worldwide, with predictions foreseeing the percentage of people aged 60 years or older to increase from 900 million in 2015 up to 2.1 billion by 2050 [1]. An important challenge of contemporary neuroscience is to better understand the systemic effects of ageing on the structure and function of the human brain [2–5]. Ageing is a major risk factor for late-onset brain disorders, accelerates cognitive and motor decline and worsens the quality of life. A deeper understanding of this process could motivate novel interventions or protection therapies against age-related deterioration or neurodegenerative diseases [6–10].

Several effects of ageing on brain structure have been previously studied. Along the lifespan, the total brain volume increases from childhood to adolescence by approximately 25% on average, remains constant for the three following decades, and finally decays back to childhood size at late ages [11]. Additionally, it has been shown that the amount of atrophy in aged brains is not homogeneous, but some anatomical regions are more affected than others—well-known ageing-targeted structures are the hippocampus [12], prefrontal cortex [13] and basal ganglia [14, 15]. White-matter degenerates faster than gray-matter along the lifespan, indicating that the overall connectivity is diminished with age [16]. Moreover, a progressive decrease in many tract-integrity measures has been shown using diffusion imaging, which is more pronounced in subjects above 60 years of age [5, 6].

Previous studies of functional connectivity along the lifespan during resting state have shown that regions within the default mode network (DMN) become less functionally

connected with age [4, 7, 17, 18]. Additionally, the frontoparietal, dorsal attention, and salience networks also show some degree of age-related decline, including reduced within-network connectivity [17, 19–21]. In contrast, between-network connectivity increase with age between the DMN, somatosensory, and the frontoparietal control networks. A stronger connectivity between the frontoparietal and dorsal attention networks has been reported as well [3, 19, 22, 23]. Furthermore, recent articles on high-order functional interactions in fMRI brain signals have shown the predominance of redundant interdependencies in older adults, consistently across all scales [24, 25]. More precisely, in [24], we investigated how high-order statistical interdependencies were affected by age using fMRI data. Our results showed that older subjects had a higher prevalence of redundant dependencies than younger subjects; moreover, this effect occurred in all orders of interaction and was shown to be highly heterogeneous across brain regions, suggesting the existence of a "redundancy core" formed by the prefrontal and motor cortices.

Taken together, these findings suggest a general loss of functional specificity or circuit segregation across brain circuits [26]. Moreover, an increase in between-network interactions is a dominant feature of aging brains. Albeit important progress in our understanding of the effects of ageing on brain function, these effects are less well understood than the effects on structural connectivity, which shows a progressive age-related disconnection.

Despite these considerable advances in understanding how the anatomical and functional connectivity change along the lifespan, the relationship between changes in brain structure and function leading to age-related decline remains largely unknown. To bridge this important gap, we sought to investigate how age-related changes in brain structure affect high-order interactions in brain function. We tackle this fundamental question through whole-brain computational modeling, which is an powerful emerging tool for investigating the neurobiological mechanisms underlying macroscale neural phenomena [27–29].

Our approach is based on the Dynamic Mean Field (DMF) model, which simulates mesoscale neural dynamics using coupled stochastic differential equations incorporating realistic neurophysiological aspects [30–32]. DMF modeling can be used to systematically perturb connectome characteristics while assessing the resulting effects on macroscale brain dynamics and function, thus opening the way to provide mechanistic modeling interpretations to data obtained from lesion studies and ageing [33–35]. Furthermore, different DMF inputs and biophysical parameters can be systematically altered in ways that are beyond the capabilities of current experimental research, which makes whole-brain computational modeling a privileged tool for investigating causal mechanisms that drive the organization and functioning of the brain [28, 36–38].

Capitalising on this powerful approach, here we employ the multimodal neuroimaging-built DMF model—including empirical BOLD fMRI dynamics and anatomical connectivity obtained from diffusion MRI. In particular, we use the DMF model to investigate how aging-induced changes in the human connectome shape high-order brain interdependencies. Following the methodology presented in Ref. [39], our analysis focuses on two types of qualitatively distinct high-order interactions which are critical in our approach: synergy and redundancy. The latter is an extension of the standard notion of correlation that applies to three or more variables, in which each variable has a 'copy' of some common information shared with other variables [39]. An example of extreme redundancy is full synchronisation, where the value of one signal makes it possible to predict the state of any other. In contrast, synergy refers here to statistical relationships that regulate a group of variables but do not constrain any subset of them [40–42]. Synergy allows for a counterintuitive coexistence of local independence and global cohesion, proposed to be instrumental for high-order brain functions, while redundancy—including instances of high synchronization, such as deep sleep or

epileptic seizures—would make neural systems less suitable for them [43–46]. Our results show that DMF reproduces age-related changes in redundancy and synergy, previously reported in Ref. [24], and that these changes are driven by a process of connectome neurodegeneration. Leveraging this evidence, we developed a non-linear neurodegenerative model that efficiently mimics connectome ageing and its associated changes in the pattern of high-order interactions of the human brain.

## Results

### The DMF model reproduces empirical differences between age groups in the high-order redundant and synergistic interactions

Our analyses are based on the DMF model, which uses structural and functional connectivity matrices, respectively SC and FC, to simulate the activity of various brain regions wired with SC in presence of local excitatory and inhibitory neuronal populations. Our analyses are based on the DMF model, which simulates the activity of various brain regions containing local excitatory and inhibitory neuronal populations, and connections between the excitatory populations that follow an anatomical connectivity matrix (SC). A biophysical haemodynamic model [47] is then used to transform the DMF model's firing rate dynamics into BOLD-like signals. The DMF is calibrated by optimising a free parameter, denoted by $G$, that allows the model to best approximate the pairwise FC matrix [28]. This procedure is illustrated in Fig 1, and details are provided in methods.

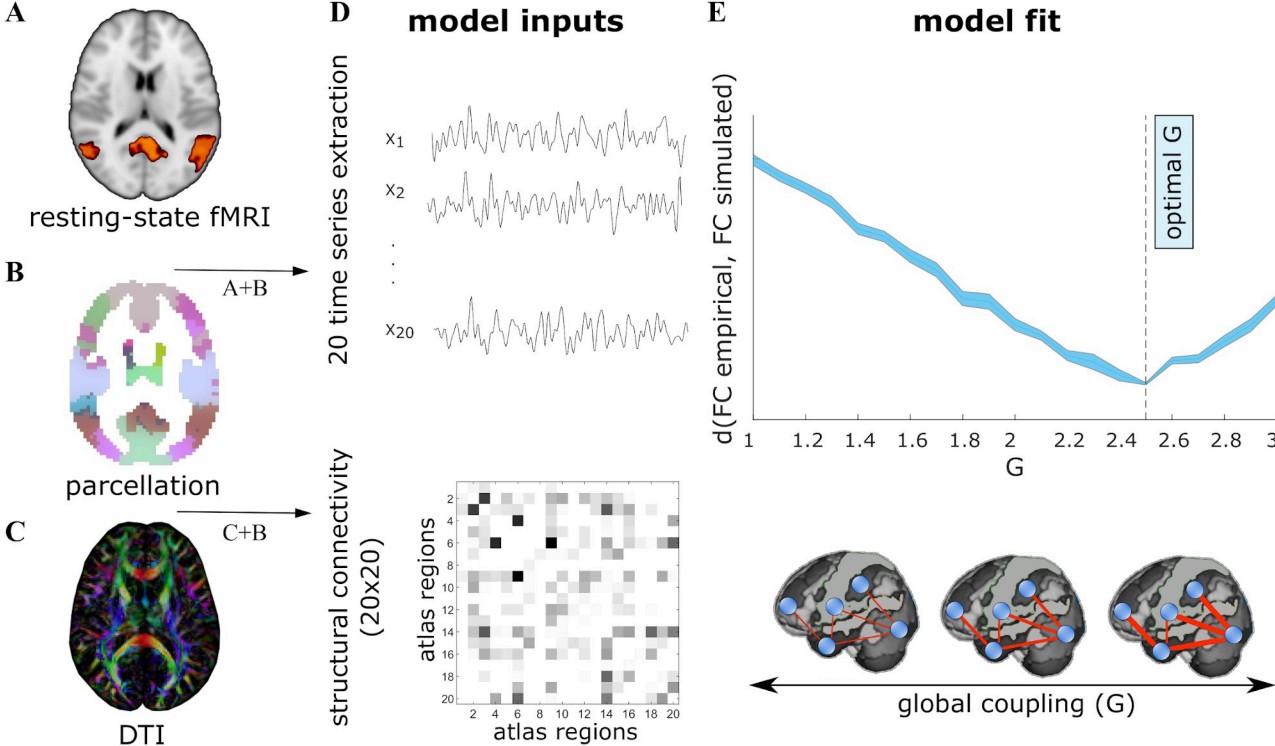

**Fig 1. Whole-brain dynamic mean field model.** The DMF model has several inputs. **A**: BOLD signals from fMRI data, **B**: a parcellation, in our case comprising 20 atlas regions, and **C**: the individual connectome obtained from DTI. **D**: Applying the parcellation to the fMRI and DTI datasets, we obtained 20 time-series signals and a $20 \times 20$ matrix representing the connectome. **E**: BOLD-like signals are simulated using the connectome and different values of the coupling parameter $G$. For each $G$, we compare the simulated and empirical data using the Kolmogorov-Smirnov distance ($d$) over the distribution of values of the FC matrices. Finally, the selected optimal $G$ is the value that minimizes this distance.

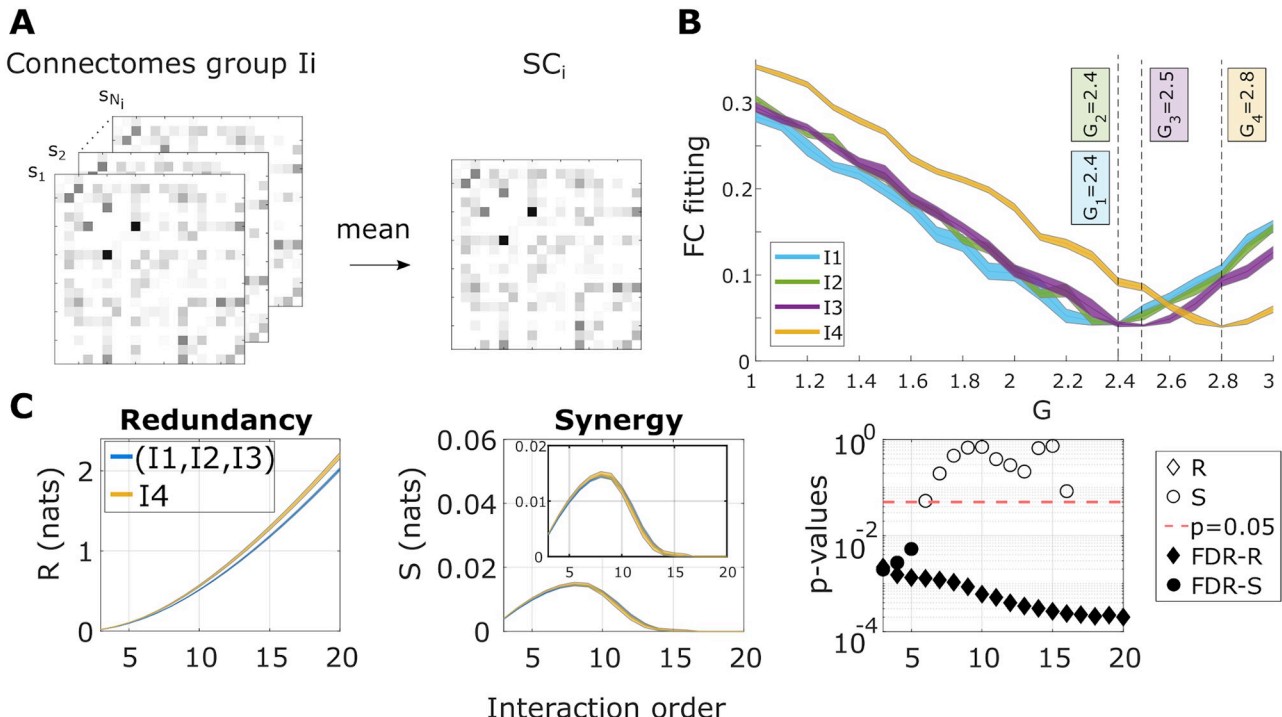

**Fig 2. The DMF model reproduces the statistically significant differences in the high-order interactions of redundancy. A**: First, the within age-group average SC matrix was calculated. **B**: Next, the optimal $G$ value for each group was obtained. **C**: We simulated brain activity within each group using the average SC and the corresponding optimal $G_i$, computed the O-information as a function of the interaction order, and separated sets of elements into the dominantly redundant (positive O-information values) and synergistic (negative O-information values). Here, the total redundancy (R) and synergy (S) was obtained as the average O-information over the redundant and synergistic sets, respectively. The p-values of the Wilcoxon rank-sum test are also depicted as a function of the interaction order, after comparing the values in I4 versus the ones obtained from the combination of (I1, I2, and I3). When the value of redundancy (or synergy) survived the false discovery rate (FDR) correction, the diamonds (or circles) were filled.

To study the effect of ageing on brain dynamics, we used functional data from different participants divided into four age groups, similar to previous work [24]: I1 ($N_1$ = 28 participants, age 10–20 years), I2 ($N_2$ = 46, 20–40 years), I3 ($N_3$ = 29, 40–60 years) and I4 ($N_4$ = 58, 60–80 years). One DMF model was built for each age group, using the average SC within each group (Fig 2A). Each model was then calibrated separately, resulting in a value of $G$ per group (Fig 2B). For each model, we simulated the brain activity using different random seeds, and the high-order interdependencies were calculated from these simulated data. In particular, we calculated the O-information [39], which can be considered an extension of the neural complexity previously proposed in [48] under the light of Partial Information Decomposition [49]. In essence, the O-information captures the balance between redundancies and synergies in a set of interacting variables [50, 51] (for further details see Methods).

We computed the O-information for all the subsets of brain regions of size $3 \geq n \geq 20$, where $n$ represents the interaction order. For each order, $n$-plets with positive and negative values of the O-information—called for simplicity 'redundancy' and 'synergy,' respectively—were calculated. Wilcoxon tests were performed to compare the average values of redundancy and synergy in the older participants (I4) with the values obtained from the groups of younger participants following previous work [24]. This is illustrated in Fig 2C. The DMF model reproduced the age differences in redundancy and synergy reported in [24]. Moreover, the redundancy differences between I4 and the rest of groups (I1, I2, I3) are statistically significant at all

interaction orders after FDR multiple comparison correction with Wilcoxon Rank-Sum (RS) statistic values RS = 29556 ($p_{FDR} < 0.001$) and RS = 28775 ($p_{FDR} < 0.005$).

Interestingly, although the DMF model was fitted only using pairwise FC values, the simulated dynamics captures that the redundancy differences between the oldest group and the rest of the subjects were significant, and this occurred for all interaction orders.

## A connectome-based model of brain ageing

Motivated by the fact that the DMF model (connected with the average SC within each age group) reproduced significant differences in high-order interactions between the groups of older people and the rest of the participants, we asked whether varying the SC of the young population was sufficient to reproduce these high-order functional characteristics in the older participants. We then studied the relationship between the weights of SC from the youngest group I1 and the corresponding weights from the oldest group I4 through a parabolic fitting (for further details see Methods). This second-order polynomial fitting revealed a non-linear dependence of the anatomical weights throughout the brain, in agreement with previous work [5, 17, 19] (Fig 3A). Next, the fitted polynomial was used to simulate the effects of ageing in each of the young participants belonging to I1, thus resulting in "synthetic aged" versions of their connectomes (Fig 3B). Each of these *synthetic* aged connectomes was used to run a set of simulations using the DMF with the optimal value $G_4$. Finally, the high-order interactions of the simulated time-courses were calculated via separating the O-information into redundancy and synergy terms. Our results, illustrated in Fig 3C, showed that the synthetically simulated

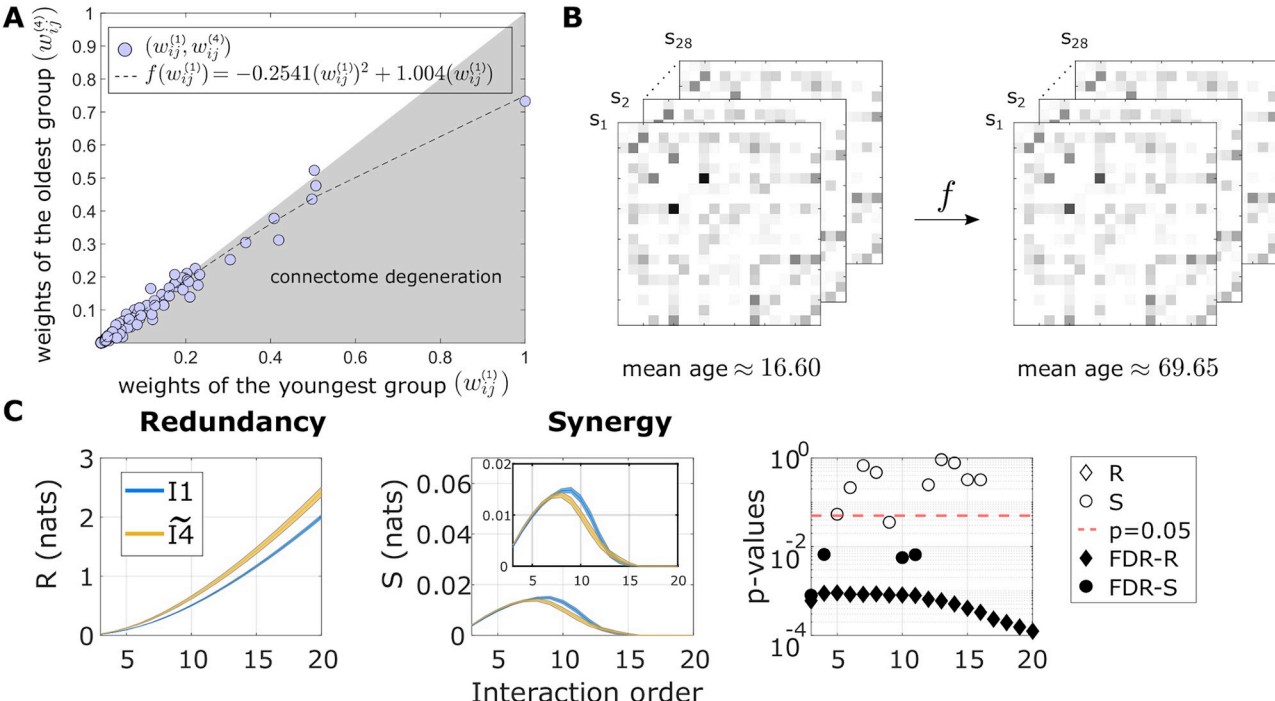

**Fig 3. Connectome-based ageing model. A**: A polynomial fit of second degree was used to link the weights of the average connectome within I1, denoted by $w_{ij}^{(1)}$, and the corresponding ones in I4 ($w_{ij}^{(4)}$). **B**: (left) Twenty-eight empirical young connectomes are transformed by the second-order polynomial fit, obtaining the synthetic aged connectomes (right). **C**: We simulated the DMF model of the aged connectomes and the optimal value $G_4$. The O-information was assessed and separated into the total redundancy (left) and synergy (center). The third panel at the right corresponds to the p-values of the Wilcoxon rank-sum test after comparing the redundancy and synergy of the synthetic I4 group with the ones in I1. When the value of redundancy or synergy survived multiple-comparison correction, both the diamonds and circles were filled.

aged participants also reproduced the functional changes observed empirically, exhibiting significant (FDR-corrected) increased redundancy at all orders, with statistic values ranging from RS = 14462 ($p_{FDR}$ < 0.001) to RS = 14264 ($p_{FDR}$ < 0.001). Moreover, and to ensure that the good performance was not the result of simply taking $G_4$ into the DMF model of the younger group, the same analysis was repeated using the linear (rather than quadratic) model of connectome degeneration (S1 Fig). The linear ageing model did not reproduce the redundancy differences observed between age groups, as they were not significant in none of the interaction orders studied. These results confirm that nonlinear heterogeneity in age-related connectome degeneration is crucial in explaining the observed changes in higher-order functional statistics. In the next section we delve into the topological structure of these anatomical changes.

## Connectome degeneration heterogeneity revealed two major differentiated communities of age-related links

We have shown that our model of connectome degeneration based on a second degree polynomial reproduces the statistically significant differences in the redundancy between the oldest group and the rest in all interaction orders studied. Interestingly, non-linearity implies that not all the links in the connectome age in the same way.

To further investigate the effects of age across brain regions, we evaluated the association between SC weights and age in all participants in our cohort (N = 161), calculating the Spearman's correlation $r$ between the participant age and each link of the SC matrix. This is illustrated in Fig 4A. To study the aging process at the aggregate or module level (beyond the

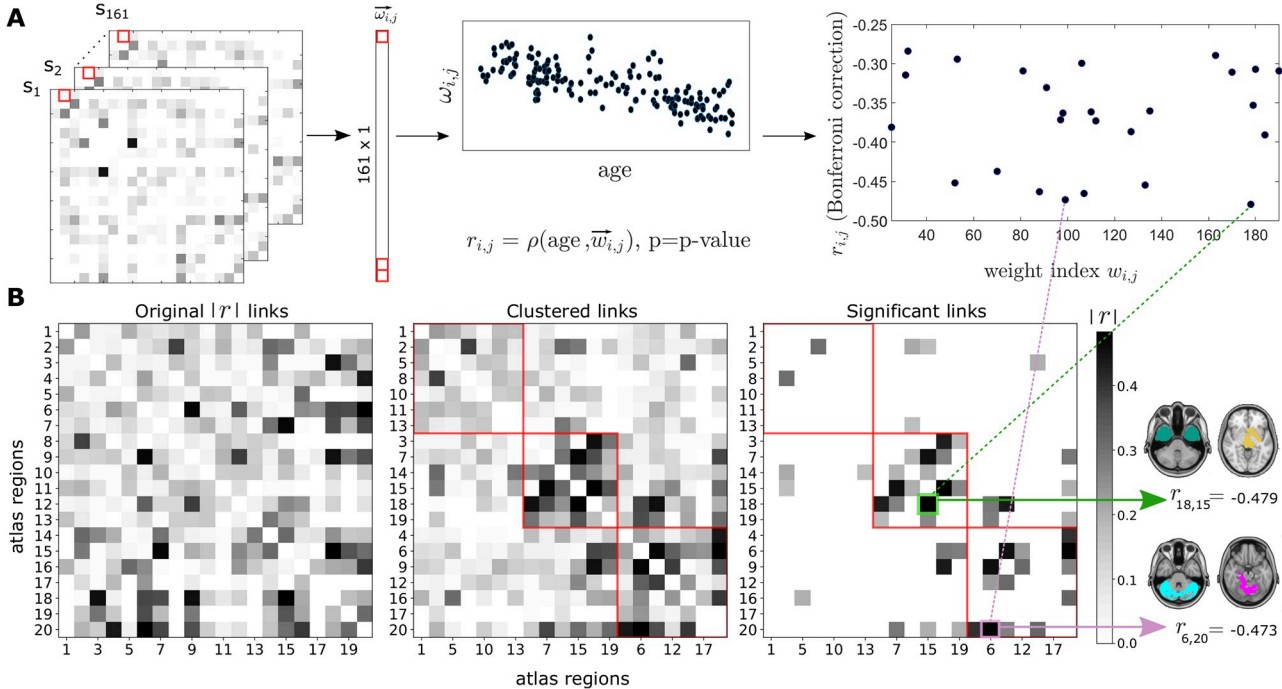

**Fig 4. Heterogeneity of the connectome degeneration. A**: The non-parametric Spearman's rank-correlation $r$ between age and individual weights $w_{ij}$ of the SC matrix was calculated across all different participants (N = 161). The final number of weights that survived to multiple comparisons is represented in the right panel, with values ranging from -0.25 to -0.50. **B**: We built a new connectivity matrix using as links the absolute values of $r$ obtained for each weight (left panel). Next, the Leuven community detection method was applied before correcting for multiple comparison and three main communities (center) were found. The links that belong to each community and that survived Bonferroni correction are also shown (right). As an illustration, we show one arbitrary link within each of these communities (colored in green and pink). Here, $|r|$ denotes the absolute values of all elements of matrix r.

relationship between weight with age at the single link level), using the absolute correlation values as links of a new matrix (left panel in Fig 4B), we applied the Louvain's community detection method obtaining three distinct link communities (note that community detection was applied to the matrix of *r* values before the multiple comparison correction). In addition, the different links within the three communities that survived the Bonferroni correction have values of *r* ranging from -0.25 to -0.5, thus showing a reduction in SC values with age. Regarding the stability of the Louvain's method to maximize network modularity, we varied the resolution parameter $\gamma$ (see Methods) and obtained the same three communities for values of $\gamma$ ranging between 0.9 and 1.2, which shows the stability of our network partition in the three communities found.

The first community only had two significant links (right panel in Fig 4B), and for this reason we considered it as a less relevant community. The second community was dominated by interactions involving brain atlas regions 15 and 18 (S2 Fig), as these two nodes had the highest values of node strength (here, calculated over the graph of r absolute values). These regions encompass several subcortical structures, such as the striatum, thalamus, brain stem, amygdala and the hippocampus (for a complete description of all regions in the atlas, see Refs. [52, 53]). In contrast, the third community, with higher strength values for regions 6,9 and 20 of the atlas, was dominated by a common structure present in these three regions, which is the cerebellum. Therefore, the two communities exhibited age-induced reduction in within-community correlations, but in one, the degeneration was centered around striatum and hippocampal connectivity, and in the other around the cerebellum.

## Discussion

In this article we used a combination of functional and diffusion MRI data together with DMF whole-brain modeling to investigate the mechanisms underlying age-variations in the structure of high-order functional interactions. The DMF model successfully reproduced the increased redundancy-dominated interdependencies of BOLD activity across brain areas in the older participants, and across all interaction orders, in full agreement with recent observations Ref. [24]. Furthermore, we provided evidence that these high-order functional changes are driven by localised non-lineal processes of neurodegeneration in the connectome. Leveraging this finding, we proposed a non-linear connectome-based degeneration model of ageing, which can be applied to young connectomes to simulate age-induced changes in functional brain patterns.

Whole-brain models of neuronal activity have significantly increased our understanding of how functional brain states emerge from their underlying structural substrate, and have provided new mechanistic insights into how brain function is affected when other factors are altered such as neuromodulation [28, 54, 55], connectome disruption [38, 56], or external stimuli [57, 58]. Adding to these findings, the present results provide a causal link between a localised connectome-based degeneration model of aging and age-variations of high-order functional interdependencies. These results establish a first step towards explaining how the reconfiguration of brain activity along the lifespan intertwines with changes in the underlying neuroanatomy.

Our results revealed two major communities with differentiated age-induced deteriorated connectivity, one focused on the striatum and hippocampus, and the other on the cerebellum. In relation to striatal connectivity, previous studies showed that the fronto-striato-thalamic circuit was the most dominant for age prediction in healthy participants [59]. Moreover, age-related deterioration of striatal connectivity has also been associated with reduced performance in rest [26] and action selection tasks [60], inhibitory control [61], and executive

function [62]. In relation to hippocampus, a gold-standard structure affecting memory-impaired degenerative diseases, is also affected in normal aging [63], with implications in spatial and episodic memories processing [64]. In relation to cerebellar connectivity, both sensori-motor and cognitive task performance in the older population has been shown to be associated with cerebellar engagement with the default mode network and striatal pathways [65]. The connectivity between cerebellum and striatum was also shown to be affected by age and exhibited relations with motor and cognitive performance [66]. Therefore, our results provide further support for the important behavioral implications that age-disconnection has on these circuits.

## Limitations and future work

This work makes use of a brain parcellation of only 20 regions from the brain hierarchical atlas (BHA) [52]. It has to be noted that, compared to other parcellations that focus into the cerebral cortex, the BHA encompasses the whole brain including brainstem, cerebellum, thalamus, striatum, amygdala, hippocampus, and cerebral cortex. While this parcellation was shown to maximize the cross modularity index between the functional and structural data, future work may also consider other brain parcellations to elucidate the robustness of our results by studying if age related changes in SC can also explain the differences high-order functional interactions in whole brain models. Analogously, some variations in the MRI preprocessing pipeline could also affects our results [67], as previous works have shown that affect pairwise FC studies [68].

In this article, we have fit the DMF model based on pairwise functional connectivity, reproducing the findings observed in Ref. [24] that the brain dynamics of older participants were significantly more redundant than the rest of the participants at all interaction orders studied. Furthermore, the use of quadratic connectome degeneration dynamics in addition to the pairwise-fitted FC model was sufficient to reproduce these high-order group differences, as pairwise-fitting with linear aging dynamics did not reproduce these results. However, other findings were also reported in Ref. [24], such as the existence of a redundant core specific to the older subject group that cannot be reproduced by fitting only pairwise functional connectivity (results not shown). Several causes could explain this; for instance, we have an average connectome per age group, which provides a global parameter $G$ per group used in all our simulations. It is possible that fitting a different model for each participant could introduce more heterogeneity, which could perhaps help improve the precise match between the actual data results and our modeling approach. A radically different alternative could have been to build models by fitting the structure of higher-order functional interactions beyond pairs, which is of great interest to explore in future work, likely providing a better fit to the data and perhaps also allowing the model to makes novel predictions, thus opening up new and exciting possibilities.

Finally, our analyses assessing high-order functional interactions are based on some specific metrics such as mean values of O-information at each interaction order. Future studies may consider different algebraic or topological properties of the full O-information hypergraph [69–74], which may provide complementary insights. It is also worth noting that the reported values of the O-information are not indicative of 'pure' synergy or redundancy, but correspond to the balance between them. The O-information was chosen because it is a convenient measure to assess high-order effects up to relatively high orders. However, the O-information is a whole-minus-sum type of measure, and hence its analysis does not fully discriminate e.g. net increases in redundancy from decreases in synergy. Future studies could perform more detailed analyses by employing partial information decomposition (PID) measures [75–82].

### Final remarks

In summary, our results extend previous findings on high-order interdependencies of the ageing brain using a novel framework that incorporated whole brain modeling and connectome datasets along the lifespan. Whole-brain models have enhanced our understanding of the brain across different conditions, and therefore provide a highly promising avenue of research in the field of ageing neuroscience. In this context, our work constitutes the first step towards mechanistic explanations on how functional high-order interdependencies in the human brain are affected by the age-connnectome degeneration. Future work should validate our modeling approach in the presence of other forms of brain degeneration, such as the interplay between aging and pathologies like Alzheimer or Parkinson diseases.

## Materials and methods

### Ethics statement

The study was approved by the local Medical Ethics Committee of KU Leuven (study number: S60428) in accordance with the Declaration of Helsinki and its amendments (World-Medical-Association 1964, 2008). All subjects were informed before study participation and signed the informed consent. For minors, written informed consent was provided by their legal representative.

### Participants

A cohort of $N$ = 161 healthy volunteers with an age ranging from 10 to 80 years (mean age 44.35 years, SD 22.14 years) were recruited in the vicinity of Leuven and Hasselt (Belgium) from the general population by advertisements on websites, announcements at meetings and provision of flyers at visits of organizations, and public gatherings (PI: Stephan Swinnen).

None of the participants had a history of ophthalmological, neurological, psychiatric, or cardio-vascular diseases potentially influencing imaging measures. The participants were divided into four distinct age groups: I1 consists of $N_1$ = 28 participants with ages ranging from 10–20 years, I2 of $N_2$ = 46 from 20–40 years, I3 of $N_3$ = 29 from 40–60 years and I4 of $N_4$ = 58 from 60–80 years.

### Image acquisition and preprocessing

Image acquisition was performed on a MRI Siemens 3T MAGNETOM Trio MRI scanner with a 12-channel matrix head coil. The anatomical images were acquired with a 3D magnetization prepared rapid acquisition gradient echo (MPRAGE) and the following parameters: repetition time (TR) = 2,300 ms,echo time (TE) = 2.98 ms, voxel size = $1 \times 1 \times 1.1$ mm$^3$, slice thickness = 1.1 mm, field of view (FOV) = $256 \times 240$ mm$^2$, 160 contiguous sagittal slices covering the entire brain and brainstem. The anatomical images were then used for preprocessing of the functional data, here acquired with a gradient echo-planar imaging sequence over a 10 min session using the following parameters: 200 whole-brain volumes with TR/TE = 3000/30 ms, flip angle = 90, inter-slice gap = 0.28 mm, voxel size = $2.5 \times 3 \times 2.5$ mm$^3$, $80 \times 80$ matrix, slice thickness = 2.8 mm, 50 oblique axial slices, interleaved in descending order. Functional imaging preprocessing was performed following a similar procedure to that in Ref. [25]. The preprocessing pipeline included slice-time correction, head motion artifacts removal, intensity normalization, regressing out of the average cerebrospinal fluid and average white matter signal, bandpass filtering between 0.01 and 0.08 Hz, spatial normalization to a template of voxel size of $3 \times 3 \times 3$ mm$^3$, spatial smoothing, and scrubbing. This resulted in a total of 2514 time series of fMRI BOLD signal for each participant, corresponding to the functional partition

used in Ref. [52]. Moreover, because for the calculation of high-order interactions at order $n$ we have to deal with $n$-plets of region combinations (for details see the following subsections), we reduced complexity grouping the original 2514 regions into 20 final brain atlas regions, simply by averaging the time series of all regions within a given atlas region. For this stage, we made use of the Brain Hierarchical Atlas [52], that has been previously used [83–86]. The partition of 20 regions is the one that maximized the cross-modularity, a metric that accounts for the triple optimization of the functional modularity, the structural modularity and the similarity between structural and functional regions (for details see Ref. [52]). To obtain the structural connectivity matrices, we acquired diffusion weighted single shot spin-echo echo-planar imaging (DTI SE-EPI) images with the following parameters: TR = 8,000 ms, TE = 91 ms, voxel size = $2.2 \times 2.2 \times 2.2$ mm$^3$, slice thickness = 2.2 mm, FOV = $212 \times 212$ mm$^2$, for each image, 60 contiguous sagittal slices were acquired covering the entire brain and brainstem. A total number of 64 volumes were acquired corresponding to different gradient directions with b = 1000 s/mm$^2$. One extra 3D diffusion image was acquired for b = 0 s/mm$^2$, needed for the diffusion imaging preprocessing. Although full details are given in Ref. [59], the pipeline consisted in eddy current correction, motion correction, tensor estimation per voxel, fiber assignment, and functional partition projection to the individual diffusion space. This resulted in SC matrices of dimension $2514 \times 2514$, one per participant, and each matrix entry corresponding with the number of white matter streamlines connecting that given region pair. Finally, we reduced complexity of these matrices by grouping the $2514 \times 2514$ matrix into $20 \times 20$ using the BHA, and averaging the BOLD signals of all regions within a given atlas region. The BOLD signals and connectomes used in this study are available at https://github.com/brincolab/High-Order-interactions/tree/master/dataset.

## Whole-brain dynamic mean field model

To simulate neuronal activity of each region, we used Dynamic Mean Field modeling (DMF) [28, 31]. Each brain region is modelled by interacting neural inhibitory (I) and excitatory (E) populations. DMF assumes that the the inhibitory currents $I^{(I)}$ are mediated by GABA-A receptors, and the excitatory ones $I^{(E)}$ by NMDA receptors. The connectivity between two different nodes $n$ and $p$ is given by the $C_{np}$. In this work, we took $C_{np}$ equal to the element of the structural matrix $SC_{np}$. Summarizing, neuronal activity followed:

$$
\begin{aligned}
I_n^{(E)} &= W_E I_0 + w_+ J_{\text{NMDA}} S_n^{(E)} + G J_{\text{NMDA}} \sum_{p=1}^{N} C_{np} S_p^{(E)} - J_n^{\text{FIC}} S_n^{(I)} , \\[2mm]
I_n^{(I)} &= W_I I_0 + J_{\text{NMDA}} S_n^{(E)} - S_n^{(I)} , \\[2mm]
r_n^{(E)} &= F\big(I_n^{(E)}\big) = \frac{g_E\big(I_n^{(E)} - I_{\text{thr}}^{(E)}\big)}{1 - \exp\{-d_E\, g_E\big(I_n^{(E)} - I_{\text{thr}}^{(E)}\big)\}} , \\[2mm]
r_n^{(I)} &= F\big(I_n^{(I)}\big) = \frac{g_I\big(I_n^{(I)} - I_{\text{thr}}^{(I)}\big)}{1 - \exp\{-d_I\, g_I\big(I_n^{(I)} - I_{\text{thr}}^{(I)}\big)\}} , \\[2mm]
\frac{dS_n^{(E)}(t)}{dt} &= -\frac{S_n^{(E)}}{\tau_{\text{NMDA}}} + \big(1 - S_n^{(E)}\big)\gamma r_n^{(E)} + \sigma\nu_n(t) \\[2mm]
\frac{dS_n^{(I)}(t)}{dt} &= -\frac{S_n^{(I)}}{\tau_{\text{GABA}}} + r_n^{(I)} + \sigma\nu_n(t)
\end{aligned}
\tag{1}
$$

where the synaptic gating variable of excitatory pools is denoted by $S_n^{(E)}$ and the one for

**Table 1. Dynamic Mean Field (DMF) model parameters.**

| Symbol | Parameter name | Value |
|---|---|---|
| $I_0$ | External current | 0.382 nA |
| $W_E$ | Excitatory scaling factor for $I_0$ | 1 |
| $W_I$ | Inhibitory scaling factor for $I_0$ | 0.7 |
| $w_+$ | Local excitatory recurrence | 1.4 |
| $J_{NMDA}$ | Excitatory synaptic coupling | 0.15 nA |
| $I_{thr}^{(E)}$ | Threshold for $F(I_n^{(E)})$ | 0.403 nA |
| $I_{thr}^{(I)}$ | Threshold for $F(I_n^{(I)})$ | 0.288 nA |
| $g_E$ | Gain factor of $F(I_n^{(E)})$ | 310 nC$^{-1}$ |
| $g_I$ | Gain factor of $F(I_n^{(I)})$ | 615 nC$^{-1}$ |
| $d_E$ | Shape of $F(I_n^{(E)})$ around $I_{thr}^{(E)}$ | 0.16 s |
| $d_I$ | Shape of $F(I_n^{(I)})$ around $I_{thr}^{(I)}$ | 0.087 s |
| $\gamma$ | Excitatory kinetic parameter | 0.641 |
| $\sigma$ | Amplitude of uncorrelated Gaussian noise $v_n$ | 0.01 nA |
| $\tau_{NMDA}$ | Time constant of NMDA | 100 ms |
| $\tau_{GABA}$ | Time constant of GABA | 10 ms |

inhibitory populations as $S_n^{(I)}$. The excitatory and inhibitory firing rates are denoted by $r_n^{(E)}$ and $r_n^{(I)}$ respectively. The feedback inhibitory control weight, $J_n^{FIC}$ was adjusted for each node $n$ in a way such that the firing rate of the excitatory pools $r_n^{(E)}$ remains fixed at about 3 Hz, using the linear fitting strategy proposed by Herzog and colleagues [38]. The precise values of the parameters used here are given in Table 1.

The complete DMF implementation was performed in Matlab, and is freely available at https://gitlab.com/concog/fastdmf.

**Haemodynamic model.** The excitatory firing rates $r_n^{(E)}$ were transformed into BOLD-like signals, following a well-known hemodynamic model [87]. It is assumed that an increment in the firing rate $r_n^{(E)}$ triggers a vasodilatory response $s_n$, that produces a blood inflow $f_n$, and changes the blood volume $v_n$ and the deoxyhemoglobin content $q_n$. In particular, we modeled these interactions as:

$$
\begin{aligned}
\frac{ds_n}{dt} =& \ 0.5 r_n^{(E)} + 3 - k s_n - \gamma(f_n - 1) \\[1em]
\frac{df_n}{dt} =& \ s_n \\[1em]
\tau \frac{dv_n}{dt} =& \ f_n - v_n^{\alpha^{-1}} \\[1em]
\tau \frac{dq_n}{dt} =& \ \frac{f_n(1-\rho)^{f_n^{-1}}}{\rho} - \frac{q_n v_n^{\alpha^{-1}}}{v_n}
\end{aligned}
\tag{2}
$$

where $\rho$ is the resting oxygen extraction fraction, $\tau$ is a time constant and $\alpha$ represents the resistance of the veins. The BOLD-like signal of node $n$, denoted $B_n(t)$, is a non-linear function of $q_n(t)$ (deoxyhemoglobin content) and $v_n(t)$ (blood volume), that can be written as:

$$
B_n = V_0[k_1(1 - q_n) + k_2(1 - q_n/v_n) + k_3(1 - v_n)]
\tag{3}
$$

where $V_0$ represent the fraction of venous blood (deoxygenated) in the resting-state, and $k_1 = 2.77$, $k_2 = 0.2$, $k_3 = 0.5$ are kinetic constants, chosen from [87].

The numerical integration of the system in Eq (2) was performed using Euler method, using an integration step of 1 ms. The signals were finally band-pass filtered between 0.01 and 0.1 Hz with a 3$^{rd}$-order Bessel filter. To match the duration of the BOLD signals obtained from the participants of this study, we simulate 160 time-points of BOLD signals corresponding to 8 minutes for each brain region. These BOLD-like signals were the ones used for the calculation of the FC matrices and the O-information, the latter used for the calculation of high-order synergistic and redundant interactions.

## Model fitting across age groups

We ran the DMF model using the average connectome $SC_i$ per age group, and chose the corresponding global coupling parameter $G_i$ by minimizing the Kolmogorov–Smirnov distance between the two distributions of FC, one obtained from simulations and the other corresponding to the empirical FC (the one obtained from the real functional data). In both cases, we built FC matrices by calculating the mutual information between pairs of time series using Gaussian Copulas. For choosing the minimizing G, we varied it from 1 to 3 with steps of size 0.1. For each $G$ value, we run 112 simulations using different random seeds. We obtained a convex curve where the x-axis represents the $G$ values and the y-axis is the Kolmogorov–Smirnov distance. The value of $G$ corresponding to the minimum Kolmogorov-Smirnov distance between the real and simulated data represents the optimal model.

## Connectome-based ageing model

We analysed the dependencies between the average SC from group I4 vs the group I1. The best fitting was achieved by a second-degree polynomial $f$, which was used as the connectome-based model for aging. For each participant of the youngest group, we modeled a synthetic *aged* version of his connectome. To obtain this synthetic connectome, we applied $f$ for each element of the matrix SC of each participant within the group I1. Finally, we used these synthetic anatomical connectivity matrices and the parameter $G_4 = 2.8$ as inputs for the DMF. For each synthetic SC, we run a number of four different simulations and obtained for each one a total number of 112 whole-brain simulations, corresponding to different initial conditions in the simulations.

Just for the results presented in S1 Fig, the aging of the connectome was done following an aging linear process. Similarly, and just for the results in S3 Fig, the connectomes of groups I2 and I3 were also quadratically aged.

## Communities of age-related links

We first assessed the association for each individual connection and age. To do so, we calculated the non-parametric Spearman's rank-correlation between each entry of the SC matrix and age, using the different participants as observations (N = 161). To find different communities of age-related brain links, we used the Louvain community detection algorithm available in the Brain Connectivity Toolbox [88] applied to the matrix of absolute values of the Spearman's $r$ values. The optimal partition of the network corresponds to a subdivision of groups of nodes (or communities) that do not overlap and that maximize the modularity of the network, a metric that increases or decreases respectively when intra-community connectivity is high or intercommunity connectivity is low. To check the stability of the communities found by the Louvain method, we vary the resolution parameter $\gamma$ (a hyperparameter that controls the trade-off between the actual number of edges in a community and the expected number of

  

edges in the same community), and compute the partition integrity coefficient by evaluating the Spearman rank correlation between the two solutions for $\gamma = 1$ and the corresponding one for each of the different simulated values of $\gamma$. The network nodes were the 20 brain regions of the Brain Hierarchical Atlas [52], and the links were the absolute value of the Spearman's correlation coefficients. After applying the community detection algorithm, we pruned the non-significant links using a Bonferroni correction.

## High-order functional interactions

Following Rosas *et al.* [89], we summarize here the main information-theoretic measures employed in this article. To begin, we defined the *total correlation* [90] TC and the *dual total correlation* [91] DTC as:

$$\mathrm{TC}(\boldsymbol{X}^n) \equiv \sum_{i=1}^{n} H(X_i) - H(\boldsymbol{X}^n) \ , \tag{4}$$

$$\mathrm{DTC}(\boldsymbol{X}^n) \equiv H(\boldsymbol{X}^n) - \sum_{i=1}^{n} H(X_i \mid \boldsymbol{X}_{-i}^n) \ , \tag{5}$$

where $H(\cdot)$ represents the Shannon entropy, and $\boldsymbol{X}_{-i}^n$ represents the vector of $n-1$ variables composed by all vector components except $X_i$ i.e., $(X_1, \ldots, X_{i-1}, X_{i+1}, \ldots, X_n)$. Both TC and DTC are non-negative generalizations of mutual information, meaning they are zero if and only if all variables $X_1, \ldots, X_n$ are statistically independent of one another.

For a set of $n$ random variables $\boldsymbol{X}^n = (X_1, \ldots, X_n)$, the O-information [89] (denoted by $\Omega$) was calculated as follows:

$$\Omega(\boldsymbol{X}^n) = \mathrm{TC}(\boldsymbol{X}^n) - \mathrm{DTC}(\boldsymbol{X}^n) \ , \tag{6}$$

The O-information is a real-valued measure that captures the balance between redundancies and synergies in arbitrary sets of variables, thus extending the properties of the interaction information of three variables [92] to larger sets (see related discussion in Ref. [49]). In particular, the sign of the O-information serves to discriminate between redundant and synergistic groups of random variables: $\Omega > 0$ corresponds to redundancy-dominated interdependencies, while $\Omega < 0$ characterizes synergy-dominated variables.

From the time series of $n$ different brain regions, we built vectors $\boldsymbol{X}^n$ and computed all these quantities using Gaussian Copulas [93]. This approach exploits the fact that the Mutual Information does not depend on the marginal distributions, and therefore, the different quantities can be conveniently transformed to Gaussian random variables from which efficient parametric estimates of high-order interactions exist. All these quantities are always computed using natural logarithms.

## Interaction order, redundancy and synergy

We calculated the O-information for all the different $n$-plets of brain regions, with $3 \geq n \geq 20$. Per participant, we computed the average O-information in which the region $m$ participates when interacting with other $n$ regions as

$$\Omega_n^m(k) = \frac{1}{\mathcal{Z}_n} \sum_{i_1} \cdots \sum_{i_{n-1}} \Omega_{(k)}(X_m, X_{i_1}, \ldots, X_{i_{n-1}}) \ . \tag{7}$$

Above, $k$ is the participant's index, $m$ the interacting region index, $n$ the interaction order, and

$$\mathscr{Z}_n \equiv \binom{M-1}{n-1}$$

is the total number of subsets of size $n-1$ in a brain partition of $M$ regions (in this work we used $M = 20$). The summations in Eq (7) included all the $n$-plets that included $X_m$. Finally, the grand average O-information of order $n$ is calculated averaging over all $M$ regions.

$$\Omega_n(k) = \frac{1}{M}\sum_{m=1}^{M}\Omega_n^m(k) \ , \tag{8}$$

We then split the O-information on positive and negative values using

$$\Omega^+ = \max\{\Omega, 0\} \quad ; \quad \Omega^- = -\min\{\Omega, 0\} \ , \tag{9}$$

so that $\Omega = \Omega^+ - \Omega^-$. Using these quantities, we calculated the following metrics for redundancy and synergy, for each subject $k$, interacting region $m$, and interaction order $n$:

$$R_n^m(k) = \frac{1}{\mathcal{N}_{n,m}^+}\sum_{i_1}\cdots\sum_{i_{n-1}}\Omega_{(k)}^+(X_m, X_{i_1}, \ldots, X_{i_{n-1}}) \ , \tag{10}$$

$$S_n^m(k) = \frac{1}{\mathcal{N}_{n,m}^-}\sum_{i_1}\cdots\sum_{i_{n-1}}\Omega_{(k)}^-(X_m, X_{i_1}, \ldots, X_{i_{n-1}}) \ , \tag{11}$$

where $\mathcal{N}_{n,m}^+$ and $\mathcal{N}_{n,m}^-$ represent the number of n-plets with positive and negative O-information values, respectively.

Finally, the average of these quantities over all subjects and regions can be also calculated as

$$R_n(k) = \frac{1}{M}\sum_{m=1}^{M}R_n^m(k) \ , \tag{12}$$

$$S_n(k) = \frac{1}{M}\sum_{m=1}^{M}S_n^m(k) \ . \tag{13}$$

The code to compute all these metrics is available at https://github.com/brincolab/High-Order-interactions.

## Statistical analyses

We used a non-parametric statistical Wilcoxon rank-sum (RS) test to compare the redundancy and synergy between simulated data of the group I4 versus simulated data of the youngest groups I1, I2, and I3 as shown in Fig 2, and on the other hand, the youngest group I1 versus the synthetic I4 group, as shown in Fig 3. We corrected the significance values for hypothesis testing for multiple comparisons by controlling the false discovery rate (FDR) following a standard Benjamini-Hochberg procedure [94].

## Supporting information

**S1 Fig. The linear aging model does not provide significant differences in redundancy between age groups.** Similar to Fig 3C in the main text, but for a synthetic I4 age group using a linear model ($\tilde{\text{I}}4_L$) for ageing instead of a quadratic one. Redundancy (left) and synergy

(middle) are plotted as a function of interaction order and then, for each interaction order, Wilcoxon rank-sum test p-values are obtained after comparing their values between the groups $\tilde{I}4_L$ and I1. If the p-values survived the FDR multiple comparison correction, both the diamonds and the circles were filled in. In this case, none of the values passed the test.
(TIF)

**S2 Fig. Node strength values across different nodes communities (C) affected by age.** Nodes here are atlas regions. **A**: Rate degeneration matrix calculated using the absolute values of *r*, and where all links in the matrix survived to Bonferroni correction (a similar matrix was shown in the right panel of Fig 4B of the main manuscript). **B**: Strength values for all atlas regions (nodes) calculated on the rate degeneration matrix show in panel A. Node strength was calculated by summing over all positive values in each row or column in the matrix. **A,B**: Colors blue (C1), orange (C2) and black (C3) indicate different communities.
(TIF)

**S3 Fig. Comparisons of redundancy and synergy when aging groups I2 and I3.** Similar to Fig 3C, here we compare redundancy and synergy by synthetic aging of the I2 and I3 groups using the quadratic aging model, giving respectively the aged groups $\tilde{I}4_2$ (panel A) and $\tilde{I}4_3$ (panel B). When comparing redundancy and synergy of these two groups with group I1, both situations provided significant redundancy differences after FDR multiple comparisons (filled diamonds) across all interaction orders.
(TIF)

# Author Contributions

**Conceptualization:** Marilyn Gatica, Fernando E. Rosas, Pedro A. M. Mediano, Patricio Orio, Rodrigo Cofré, Jesus M. Cortes.

**Data curation:** Marilyn Gatica, Ibai Diez, Jesus M. Cortes.

**Formal analysis:** Marilyn Gatica, Rodrigo Cofré.

**Investigation:** Marilyn Gatica.

**Methodology:** Marilyn Gatica, Fernando E. Rosas, Pedro A. M. Mediano, Patricio Orio, Rodrigo Cofré, Jesus M. Cortes.

**Project administration:** Rodrigo Cofré, Jesus M. Cortes.

**Resources:** Stephan P. Swinnen.

**Supervision:** Fernando E. Rosas, Pedro A. M. Mediano, Rodrigo Cofré.

**Visualization:** Marilyn Gatica.

**Writing – original draft:** Marilyn Gatica, Fernando E. Rosas, Pedro A. M. Mediano, Stephan P. Swinnen, Rodrigo Cofré, Jesus M. Cortes.

**Writing – review & editing:** Marilyn Gatica, Fernando E. Rosas, Pedro A. M. Mediano, Stephan P. Swinnen, Rodrigo Cofré, Jesus M. Cortes.

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
