## [Decision Letter · Decision Letter 0]

10 Mar 2022

Dear Dr. COFRE,

Thank you very much for submitting your manuscript "High-order functional interactions in ageing explained via alterations in the connectome in a whole-brain model" for consideration at PLOS Computational Biology.

As with all papers reviewed by the journal, your manuscript was reviewed by members of the editorial board and by several independent reviewers. In light of the reviews (below this email), we would like to invite the resubmission of a significantly-revised version that takes into account the reviewers' comments.

We cannot make any decision about publication until we have seen the revised manuscript and your response to the reviewers' comments. Your revised manuscript is also likely to be sent to reviewers for further evaluation.

Sincerely,

Demian Battaglia

Guest Editor

PLOS Computational Biology

Lyle Graham

Deputy Editor

PLOS Computational Biology

Jason A. Papin

Editor-in-Chief

PLOS Computational Biology

Reviewer's Responses to Questions

**Comments to the Authors:**

Reviewer #1: The paper " High-order functional interactions in ageing explained via alterations in the connectome in a whole-brain model" focuses on the study of the biological (or mechanistic) origin of high-order dependencies measured using informational synergy and redundancy (more precisely, the O-information framework recently proposed by Rosas et al.). In the paper a whole-brain model is built using state-of-the-art techniques using structural and functional connectivity (fMRI) data from a cohort of subjects of age varying from 10 to 80 years.

In particular, the authors fit the computational model to the standard functional connectivity matrix (FC) as it is conventionally done, and then measure the high-order properties of the resulting synthetic timeseries, uncovering a dependence of synergy and redundancy similar to that observed in data in a previous publication by some of the authors (namely a growth of redundancy with the order interactions and a peak for synergy at an intermediate order).

To show that the origin of these properties lies in the brain's structural degeneration, they aged artificially the young structural connectomes (using a quadratic model fit) and simulated again the corresponding functional connectivity, showing again similar results to the ones previously obtained for real data. Finally, motivated by the effects of structural degeneration, the authors use a standard community detection algorithm to extract communities on a new adjacency encoding the correlation between age and degeneration for each link, finding two main communities.

The paper is original and reports its central concept quite nicely. The language is clear and the explanations can be easily followed overall. I do however have several conceptual and technical concerns that I would like to see addressed before I can consider this contribution appropriate for publication in plos computational biology.

Comments:

- my first comment is conceptual: why should we fit the FC matrix alone instead of directly the set of higher-order redundancy/synergy values? Is this to state that pairwise correlations are sufficient to describe all the higher-order dependencies? But if so then, why bother with them if fixing the FC statistics is already sufficient to reconstruct the high-order ones?

- the paper states that the DMF (Figure 2, and the "aged" young connectomes in Figure 3) reproduce the observed patterns from ref 24. However, a fairer assessment would be that the synthetic time-series show a qualitatively similar behaviour to the original data, e.g. synergy has a peak for intermediate order, while redundancy keeps growing with the order. Quantitatively, however there seems to be a considerable discrepancy, e.g. the synergy peak in data is around order 12, while in simulations it's around 7; the values of redundancy and synergy too seem overall much smaller than those found in the data (e.g. synergy peaks at \\~0.05 in simulations, but above 0.1 in data) and so on. I wonder whether this is a question of different scales across the two paper, but I think it would be very important to plot the R/S values from data in Figure 2 at least, for comparison to those obtained from the synthetic time series.

- I did not find the details about the linear model to fit the structural degeneration mentioned in the main text and of the comparison of its O-information to the ones obtained by the DMF and the quadratic model.

- why after showing that a non-linear model is needed to link age and structural degeneration, a linear correlation measure was used to build the matrix used for community detection in figure 4? Wouldn't a rank-correlation be more appropriate? How would that change the reported results ?

- community detection #1: why was Louvain used? More precisely, what was the matrix given as input to the Louvain algorithm? the actual signed matrix or its positivised version? This is important, because, to my understanding, in its standard formulation (like in BCT), modularity (and the Louvain algorithm) accepts weighted graphs but it tries to maximise the sum of edge weights within a module and minimize that of links in between modules. The paper says that most edges (Fig. 4) are negative, thus the communities found should be those that have the smallest negative values inside. However, the plot reports the absolute value of r, and the block-reordered version (Fig. 4b) shows credible blocks. can you please provide a bit more detail?

- community detection #2: why modularity? Modularity always gives a result even when it's not statistically justified or below the resolution limit. Have you tried reproducing the results using different community detection algorithms (e.g. stochastic block models), which although different in nature, have also the possibility of returning more general architectures than block structures and -when appropriate--no blocks at all?

- node properties: node strength with reference to the matrix defined in Fig.4 is used multiple times in the text. How are these strengths computed (in linght of the fact that all edge weights are negative)?

- statistical comparison: multiple times in the text multiple comparisons and corresponding corrections are mentioned

but no statistics are reported for the tests, how they were performed, etc. Can you please provide additional details?

Reviewer #2: Dear Editor,

I am attaching my report on the manuscript entitled: High-order functional interactions in age ageist explained via alterations in the Connectome in a whole-brain model, by Gatica et. Al.

In this work, the authors address ageing under a high-order functional perspective. Inspired by a recent empirical result in high order interactions on functional brain networks (ref. 24, where high order metrics were empirically proven to be a biomarker of ageing), the authors developed a realistic model - whole brain dynamic mean-field model as well as Haemodynamic model - which includes both functional and structural connectomes.

By applying the model to the youngest age group (I1) and comparing it to the oldest group (I4), the model reproduced the findings in ref. 24 on high order interactions, i.e., similar age variations in high order interactions of redundancy and synergy.

The paper is well written, and the results are sound. Yet, the authors made their codes available for reproducibility purposes, and I would like to commend them for that. Ageing is a relevant topic per si, and coming with a model that reproduces high order aspects of ageing may impact the next steps of high order neuroscience.

Based on the report above, I believe that the manuscript fulfil the relevance and quality criteria for publication in PLOS computational biology. However, before publication, I think that the authors should address the following (minor) points:

1) Since the manuscript is based on the findings of ref. 24, for self-consistently purposes, I would summarize a bit more the results of ref. 24 in this manuscript. I knew ref. 24 in advance, but a reader not familiar with ref. 24 may not follow this manuscript.

2) The authors used the model to predict High order metrics of an "aged version" of group I1 and compared it with group I4. Would those findings be doable (or applicable) for intermediate groups (I2 and I3)? Since the authors mentioned future work in other forms of brain degeneration, some disorders (e.g. Multiple sclerosis) may start at ages in groups I3 (or even I2).

3) The authors assessed the relationship between each individual SC link and age. They computed the Pearson correlation between SC matrix and age to do so. However, It's known (and also mentioned in this manuscript) that multiple properties related to ageing are nonlinear and often quadratic. The authors used a quadratic fitting in Fig. 3A to model Connectome degeneration between group I1 and I4. Therefore, as a posthoc analysis, I would suggest the use of connectivity metrics that can capture eventual nonlinear relations between the SC links and age (e.g. mutual information), which could be more appropriate than Pearson correlation (which captures only linear relationships between variables).

4) The fact that the model is pairwise but still captures high order interactions is quite intriguing. It would be interesting if the authors could develop or discuss a bit more on this result. Where do the high order interdependencies come from? Are they possibly coming due to the inclusion of two connectivity modalities (SC and FC)? Or from the nonlinearities of the model?

5) There is another interesting result in ref. 24 that the authors did not report a comparison with the model developed here. The authors of ref. 24 reported the presence of a "redundancy core" in rs-fMRI and that this core changes as you age. In fact, the authors noted that the redundancy core of groups I1, I2 and I3 are different (and bigger) than group I4. Are these results also found in the model proposed here?

**Have the authors made all data and (if applicable) computational code underlying the findings in their manuscript fully available?**

Reviewer #1: Yes

Reviewer #2: Yes

PLOS authors have the option to publish the peer review history of their article (what does this mean?). If published, this will include your full peer review and any attached files.

Reviewer #1: No

Reviewer #2: No
---

## [Decision Letter · Decision Letter 1]

23 Jul 2022

Dear Dr. COFRE,

We are pleased to inform you that your manuscript 'High-order functional interactions in ageing explained via alterations in the connectome in a whole-brain model' has been provisionally accepted for publication in PLOS Computational Biology.

We also suggest that you consider Reviewer 1's comments on the title, but we leave any decision to change it up to you.

Best regards,

Lyle J. Graham

Deputy Editor

PLOS Computational Biology

Reviewer's Responses to Questions

**Comments to the Authors:**

Reviewer #1: The authors have addressed my comments. I am satisfied with the current manuscript.

My only further comment is that the title might be a little confusing.

In particular, I think the authors should consider modifying the title to explicitly include something point to the type of higher-order observables they measure, that is, synergy-redundancy.

This is important in my opinion because at the moment the term higher-order interactions is being used in many different ways (polyadic, markov-chain path networks, TDA, information theoretic) in the area of "higher-order" applications (this is something pretty much everyone in the field is to some level guilty of, me included tbh).

So it would be worth adding a word somewhere in the title specifying this work focuses on information-theoretic quantities, e.g. "High-order functional synergy/redundancy" or "High-order information explains... ".. I don't really have a good suggestion for the authors, I trust them to find an appropriate small modification.

Reviewer #2: The authors successfully answered the questions and criticisms and improved the manuscript according to the suggestions and comments I raised and, in my opinion, raised by the Second Referee. Therefore, I'm pleased to recommend the paper for publication in Plos Computational Biology.

**Have the authors made all data and (if applicable) computational code underlying the findings in their manuscript fully available?**

Reviewer #1: Yes

Reviewer #2: Yes

PLOS authors have the option to publish the peer review history of their article (what does this mean?). If published, this will include your full peer review and any attached files.

Reviewer #1: No

Reviewer #2: No

---

## [Editor Report · Acceptance letter]

29 Aug 2022

PCOMPBIOL-D-21-01766R1 

High-order functional redundancy in ageing explained via alterations in the connectome in a whole-brain model

Dear Dr COFRÉ,

I am pleased to inform you that your manuscript has been formally accepted for publication in PLOS Computational Biology. Your manuscript is now with our production department and you will be notified of the publication date in due course.

With kind regards,

Zsofia Freund
